# Identifying Risk Factors for Delirium Through Comprehensive Geriatric Assessment in Older Adults Receiving Palliative Cancer Care

**DOI:** 10.3390/nursrep15090328

**Published:** 2025-09-08

**Authors:** Paula Llisterri-Sánchez, Francisco Miguel Martínez-Arnau, Pilar Pérez-Ros

**Affiliations:** 1Doctoral School, Catholic University of Valencia San Vicente Mártir, 46001 Valencia, Spain; paula.ll.s@mail.ucv.es; 2Department of Physiotherapy, University of Valencia, Gascó Oliag 5, 46010 Valencia, Spain; francisco.m.martinez@uv.es; 3Department of Nursing, University of Valencia, Menendez Pelayo s/n, 46010 Valencia, Spain

**Keywords:** cancer, delirium, geriatric assessment, palliative care, risk factors

## Abstract

**Background/Objectives**: Older adults with cancer are at high risk of developing delirium. Comprehensive geriatric assessment (CGA) is a fundamental tool for prioritizing problems and establishing appropriate interventions in older patients. This study aimed to identify risk factors for delirium through a CGA in older adults receiving palliative cancer care in hospital. **Methods**: This longitudinal observational study included people aged 65 years or over who were hospitalized in medical wards with an advanced stage of cancer. Clinicians performed a CGA and screened for delirium using the Confusion Assessment Method (CAM). Diagnosis of delirium was based on criteria set out in the Diagnostic and Statistical Manual of Mental Disorders, Fifth Edition (DSM-V). We calculated odds ratios (ORs) with 95% confidence intervals (CIs) to compare different variables in people with versus without delirium. **Results**: The study included 105 participants, of whom 67 (63.81%) developed delirium during follow-up. The mean age was 71.33 years in the delirium group and 72.24 years in the non-delirium group. Risk factors for delirium were dysphagia (OR 2.45, 95% CI 1.01–5.99; *p* = 0.045), urinary catheterization (OR 2.97, 95% CI 1.09–8.13; *p* = 0.029), and having at least one delirium episode in the last year (OR 5.68, 95% CI 1.97–16.34; *p* = 0.001). The predictive model showed that older male cancer patients with a urinary catheter and dysphagia are most likely to develop delirium in hospital (area under the curve [AUC] = 0.679, 95% CI 0.577–0.780; *p* = 0.002). **Conclusions**: The prevention and effective management of delirium require a person-centered, interdisciplinary approach that considers both clinical and psychosocial aspects. Including variables such as male sex, dysphagia, and urinary catheterization in delirium risk assessment enables more comprehensive and personalized management.

## 1. Introduction

Cancer prevalence increases significantly with age, primarily due to accumulated cellular damage from lifelong risk factor exposure and the declining effectiveness of cellular repair mechanisms. Cancer is associated with higher morbidity and is the second leading cause of death in older people [1].

Mental disorders, particularly delirium and anxiety, are very common in older people with cancer [2]. Delirium is an acute neuropsychiatric syndrome characterized by alterations in attention, consciousness, and cognition, with a fluctuating course [3]. Older people are more vulnerable to the onset of delirium during hospitalization, especially those with cancer. The incidence of delirium in older cancer patients can reach 25% to 50%, with even higher rates in people receiving palliative care or with advanced stages of the disease [4].

Older adults with cancer are at high risk of developing delirium due to a combination of predisposing and precipitating factors [5]. Predisposing factors include advanced age, pre-existing cognitive impairment, multimorbidity, frailty, and dependence; while precipitating factors include the use of drugs with delirium-inducing potential (opioids, benzodiazepines, corticosteroids, anticholinergics), infections, electrolyte imbalances, uncontrolled pain, surgical interventions, prolonged immobility, and exposure to the hospital process itself [6]. In addition, cancer treatments (e.g., chemotherapy, radiotherapy) and progression of the disease can alter the patient’s metabolic and neurological status, making them even more vulnerable to delirium [7]. The interaction between adverse treatment effects and predisposing and precipitating factors further increases this risk [8].

Delirium is associated with poor health outcomes, including increased in-hospital and long-term mortality, prolonged hospital stays, higher rates of institutionalization, loss of functionality and cognition, and reduced quality of life for patients and their caregivers [9]. In older cancer patients, delirium can interfere with clinical decision-making, hinder doctor-patient communication, and limit access to appropriate treatments or palliative care. In people with advanced cancer, delirium can present as a terminal symptom, further complicating end-of-life management and planning [10].

Given the high prevalence and serious implications of delirium, it is crucial to implement strategies for the prevention, early detection, and multidisciplinary management of this syndrome in older cancer patients. Conducting a comprehensive geriatric assessment (CGA), using systematic screening scales such as the Confusion Assessment Method (CAM), and minimizing precipitating factors can help improve the prognosis and quality of care in this vulnerable population [11].

The challenges of providing palliative care for older people with cancer are well documented in the literature, as the illness trajectory and needs of this population are unique [12]. The clinical management of older patients transcends traditional paradigms, which focus primarily on diagnosing and treating isolated diseases. Instead, it involves assessing clinical complexity that spans multiple domains (including physical, cognitive, psychological, affective, socioeconomic, and environmental aspects) [13]. Geriatric healthcare teams have long used CGA to address these challenges and design personalized interventions tailored to each patient’s needs, priorities, and resources. The key components of CGA cover comorbidities, physical function, cognitive status, mood, fall risk, polypharmacy, social support, and nutrition [13].

Although previous studies have identified risk factors for delirium in people with cancer receiving palliative care, there is a lack of comprehensive analysis in older people. The objective of this study was to identify the risk factors for delirium after performing a CGA in older people diagnosed with advanced cancer and receiving palliative care.

## 2. Materials and Methods

### 2.1. Study Design, Setting and Participants

This longitudinal observational study included all people aged 65 years and older with an advanced stage of cancer who were admitted to the medical oncology unit of Valencia General University Hospital between June 2021 and December 2022 for palliative care and who provided their informed consent to participate. The clinical research ethics committee of the hospital approved the study (ref: 91/2021).

### 2.2. Sample Size

From January to December 2021, 141 people meeting our inclusion criteria (other than period of admission) were hospitalized in the oncology unit. Assuming a heterogeneity level of 50%, a 5% margin of error, and a 95% confidence interval (CI), the sample size estimated for this study was 103 participants.

### 2.3. Procedure

The study nurse provided full details of the study to potential participants and their families following admission to the oncology unit. All patients who accepted and signed the informed consent form were recruited. Socio-demographic data such as age and sex were collected. Variables collected from participants’ medical records included the main cancer diagnosis and risk factors for delirium: dementia; incontinence (and type of incontinence); urinary catheterization; urinary tract infection in the last six months [14]; kidney failure; falls in the last month; dysphagia; hospitalizations in the last six months; emergency department visits in the last six months; number of drugs prescribed daily; use of anticholinergics, anxiolytics, sedatives, opioids, antidepressants, and neuroleptics; and polypharmacy (≥5 prescribed drugs). Six months was considered an acceptable duration for evaluating the presence of urinary tract infection because older people often receive prolonged treatment for such infections.

All participants had a CGA on the first day of hospitalization, including the following dimensions:Functional status according to the Barthel Index, which evaluates independence in activities such as eating, transferring between chair and bed, personal hygiene, using the bathroom, bathing or showering, walking, going up and down stairs, dressing, and sphincter control (stool and urine). Scores assigned to each activity range from 0 (total dependence) to 10 or 15 (total independence), and the total score ranges from 0 to 100 [15].Nutritional status assessed with the Mini Nutritional Assessment (MNA), which includes 18 questions and enables quick identification and planning of nutritional interventions. The tool is based on anthropometric variables, diet, overall assessment, and self-perception of health and nutrition. It can be used in various settings (primary care, nursing homes, and hospitals). The MNA classifies individuals into three categories according to their total score: 24–30 points, normal nutritional status; 17–23.5 points, at risk of malnutrition; <17 points, malnourished [16].Cognitive status assessed with the Mini Mental State Examination (MMSE). The tool assesses six main cognitive domains: temporal and spatial orientation, immediate and delayed memory, attention and calculation, language, and visuospatial skills. The total score ranges from 0 to 30 points, and the cut-off point for dementia is usually set at under 24 points [17].Social support of the patient and their family measured on the Gijón social-familial evaluation scale (GSFES), which covers family, economic status, housing, social relationships, and support received, with scores that allow for the classification of social risk levels. This scale ranges from 0 to 25, with higher scores indicating greater social risk: 5–9 points, low social risk; 10–14 points, moderate social risk; ≥15 points, high social risk [18].

No participants had delirium at hospital admission. The nursing staff assessed participants with the CAM scale three times daily (morning, afternoon, evening) during the hospital stay. If this assessment indicated delirium, the diagnosis was confirmed against the Diagnostic and Statistical Manual of Mental Disorders, Fifth Edition (DSM-V) diagnostic criteria.

The short-form CAM evaluates four cognitive features:Acute onset and fluctuating courseInattentionDisorganized thinkingAltered level of consciousness

People who meet the criteria for features 1 and 2 as well as 3 and/or 4 are diagnosed with delirium [19]. The instrument has high sensitivity (94–100%) and specificity (90–95%) against diagnosis by a geriatrician.

The DSM-V criteria for delirium are:A.A disturbance in attention and awareness.B.The disturbance develops over a short period of time, represents a change from baseline attention and awareness, and tends to fluctuate in severity during the course of a day.C.An additional disturbance in cognition.D.The disturbances in Criteria A and C are not better explained by a preexisting, established, or evolving neurocognitive disorder and do not occur in the context of a severely reduced level of arousal, such as coma.E.There is evidence from the history, physical examination, or laboratory findings that the disturbance is a direct physiological consequence of another medical condition, substance intoxication or withdrawal, or exposure to a toxin, or is due to multiple etiologies [20].

### 2.4. Statistical Analysis

An impartial third party validated all data submitted to the database. Descriptive statistics were produced to summarize the information: relative frequencies for categorical variables (age > 75 years; sex; dementia; incontinence; urinary catheterization; UTI in last six months; kidney failure; dehydration; fall in previous 30 days; dysphagia; delirium in last 12 months; use of anticholinergics, anxiolytics, sedatives, opioids, antidepressants, and neuroleptics; polypharmacy [≥5 prescribed drugs]; and dichotomized scores of CGA tools [Barthel Index < 60 points; MNA < 17 points; MMSE < 10 points, GFES ≥ 15 points]) and mean values with standard deviations (SDs) for normally distributed continuous variables (age and scores obtained on the Barthel Index, MNA, MMSE, and GFES).Our analysis employed the odds ratio (OR) with 95% CI to measure the association between delirium and each dichotomous risk factor, and the mean difference (MD) with 95% CI for continuous risk factors.

A binary logistic regression model, adjusted for age and sex, was fitted for delirium to investigate the significance of the risk factors. The first complete model, with all the variables in the bivariate analysis, was significantly associated with delirium. In a subsequent phase, the model dropped all variables that produced no substantial change when excluded (adjusted effect ≤ 10%) and all variables that did not improve the standard error. Where two or more subsets of variables showed the same goodness-of-fit, the investigators aimed to reach consensus through discussion. The variables included in the backward Wald model were age, sex, urinary tract infection, urinary catheterization, dehydration, incontinence, falls, dysphagia, anticholinergics, anxiolytics, antidepressants, neuroleptics, sedatives, opioids, and emergency department visits.

The study data were entered into MS Excel spreadsheets and analyzed with SPSS (Version 28.0. Armonk, NY, USA: IBM Corp.).

## 3. Results

The study included 105 participants, of whom 67 (63.81%) developed delirium. The mean age of participants was 71.66 (SD 4.1) years, with no difference between the delirium group (DG) and non-delirium group (NDG). The proportion of men was significantly higher in the DG versus the NDG (*p* = 0.042). There was a higher percentage of gastrointestinal cancer and lymphoma in the DG, and a higher percentage of lung and skin cancer in the NDG (Table 1). The mean length of hospital stay was 14.12 (SD 16.1) days, and the mean duration of delirium was 6.01 (SD 4.82) days.

The CGA showed high functional dependence, with mean Barthel Index scores below 30 points in both groups. In addition, both groups had scores on the different scales indicative of malnutrition (MNA), severe cognitive impairment (MMSE), and moderate social risk (GSFES). The DG presented higher mean GSFES scores than the NDG (13.25 [SD 2.45] vs. 12.42 [SD 2.59]; *p* = 0.036) (Table 2).

In the bivariate analysis of risk factors, male sex was associated with delirium (OR 2.31, 95% CI 1.02–5.20; *p* = 0.041), but advanced age was not. Both groups had high prevalences of dementia and incontinence, and lower prevalences of kidney failure and dehydration, without significant between-group differences. Clinical variables significantly associated with delirium were dysphagia (OR 2.45, 95% CI 1.01–5.99; *p* = 0.041), urinary catheterization (OR 2.97, 95% CI 1.09–8.13; *p* = 0.024), and having at least one episode of delirium in the last 12 months (OR 5.68, 95% CI 1.97–16.34; *p* < 0.001). No pharmacotherapeutic groups were associated with delirium, and there were no statistically significant differences between groups in the dichotomized scores indicating functional, nutritional, cognitive, and social status (Table 3).

The logistic regression model was statistically significant (Chi^2^ = 12.100; *p* = 0.007). The model explained 14.9% (Nagelkerke’s R^2^) of the variance in delirium and correctly classified 65.7% of the cases. Sensitivity was 34.2% and specificity was 83.6%. The positive predictive value was 67% and the negative predictive value was 38% (Table 4).

Older men with cancer who have a urinary catheter and dysphagia are at increased risk of developing delirium in hospital (area under the curve [AUC] = 0.679, 95% CI 0.577–0.780; *p* = 0.002).

## 4. Discussion

Delirium is a high-prevalence syndrome in older people with cancer. Knowing the risk factors during hospital admission helps nursing professionals implement detection, treatment, and prevention strategies. The objective of this study was to develop a model of the predictors of delirium in older hospitalized cancer patients receiving palliative care. The results suggest men with dysphagia and a urinary catheter are the most vulnerable group.

Older people with cancer in palliative care are at higher risk for delirium due to the confluence of age- and disease-related risk factors. The incidence of delirium in the sample analyzed was 63.8%. Previous studies have suggested that between 50% and 85% of cancer patients receiving palliative care in hospital develop delirium. Variation in percentages across studies is due to the stage of the disease, aggressive treatments, infections, metabolic imbalances, and other health problems [21].

The CGA results suggested this study sample had high functional dependence, a risk of malnutrition, cognitive impairment, and moderate social risk. These characteristics, combined with the advanced stage of the disease, make care challenging and increase the likelihood of major geriatric syndromes such as delirium [22]. Therefore, identifying individuals with greater multimorbidity, vulnerability, and excessive disability is important. Functional decline is the most accurate indicator of poor outcomes and mortality, regardless of clinical diagnosis. The stage of cancer was a key factor in this case, as it is directly related to mortality [5]. In the sample analyzed, both groups had scores indicative of malnutrition. Malnutrition is highly prevalent in people receiving palliative care [23]. The use of different treatments associated with increased mucosal inflammation, loss of appetite, and increased sedentary lifestyles can lead to weight and muscle mass loss, mainly due to decreased protein intake [23,24]. In the cognitive dimension, despite the low prevalence of diagnosed dementia, both groups had MMSE scores indicative of cognitive impairment. Cancer and cancer treatments can disrupt the compensatory mechanisms that support everyday functioning as people age [25]. Another factor that could have influenced the prevalence of cognitive dysfunction in both groups is the presence of chemo-brain (thinking and memory problems that occur in cancer patients during and after chemotherapy, also known as cognitive dysfunction or chemo-fog [26]). Unfortunately, this variable was not collected. Finally, both groups reported high levels of social support. This aspect is fundamental in the older palliative care population, as it influences morbidity and mortality [27].

CGA is an essential tool in the care of older cancer patients. When implemented effectively, it can guide oncologic treatment plans and improve communication about care planning and aging-related issues. This leads to enhanced patient-centered outcomes, treatment completion, and reduced toxicity. Having a predetermined non-oncologic intervention plan (that covers possible decisions regarding diagnostic tests and desired treatment) is crucial in cases where geriatric or interdisciplinary advice is unavailable [28,29], as it focuses on each individual’s needs regardless of the presence of cancer.

Few risk factors reported in the literature were present in this study sample. Having a history of delirium was the leading risk factor, because delirium, even if transient, indicates brain vulnerability and may be a sign of disease progression or poor response to treatment. In addition, people who have experienced delirium are more likely to have other risk factors such as dementia or cognitive impairment, as observed in the sample [6]. Among the predisposing factors described in the literature, only male sex was associated with an increased risk of delirium in this study. In contrast, advanced age, dementia, and other diseases showed nonsignificant associations with delirium. The advanced stage of the disease among cancer patients receiving palliative care may diminish the impact of other factors [6,30].

Precipitating factors associated with increased risk of delirium in this study were dysphagia and urinary catheterization. Dysphagia is prevalent in people with serious and life-limiting illnesses, often profoundly impacting their quality of life and that of their caregivers. It is a common symptom at the end of life and an indicator of poor prognosis. Dysphagia can lead to aspiration, malnutrition, dehydration, pneumonia, and even death. In the sample analyzed, malnutrition may have been related to dysphagia. It is essential to recognize the signs of dysphagia and make informed decisions about nutrition, hydration, and medication, especially in people receiving palliative care. This urgency is compounded by the established link between urinary catheterization and delirium [31]. There are several reasons why urinary catheterization increases the risk of delirium: the insertion and use of a urinary catheter can cause pain, inflammation, and irritation, and increases the likelihood of infection [32].

The use of anticholinergic drugs, opioids, sedatives or benzodiazepines were not risk factors for delirium in this sample. The patients received care in a hospital setting with treatment adjustments when needed, daily or even throughout the day, which may have biased the association between the pharmacological groups [6,33].

Previous studies have reported an association between psychosocial factors and delirium, but there were no differences between groups in this study. Social isolation, anxiety, depression, and loss of family role can aggravate the condition or make it difficult to detect; however, as the patients in this study were hospitalized, both caregiver and nursing staff surveillance may have minimized the risk. This suggests studies on delirium in the home environment are warranted [34,35].

The results of this study can help identify the older population with cancer at higher risk of delirium during hospital admission, which will enable the implementation of prevention and early identification strategies.

### Limitations and Future Developments

This study has several limitations. The first is the type of sampling, since only hospitalized patients were included. It was not possible to analyze the severity or duration of delirium or the association of delirium with each type of cancer treatment, the use of chemotherapy, and the existence of possible brain metastases.

Prospective studies conducted exclusively in older populations are needed, as most studies consider adults of all ages. Studies with larger sample sizes could stratify patients by palliative care setting (home or hospital) and by cancer type and stage. Implementing CGA in clinical practice would allow for a holistic evaluation of older adults, and using validated instruments would enable effective comparison of the results.

Nursing staff are key professionals in the prevention of geriatric syndromes, particularly delirium. Awareness among nursing professionals of the need to prevent and identify delirium is essential, as they spend the most time with hospitalized patients. Research should be geared toward risk identification and early detection, as this will help in the implementation of early personalized treatment and thus minimize the consequences of delirium.

## 5. Conclusions

Comprehensive geriatric assessment (CGA) is a tool designed to evaluate older adults in all dimensions and thus identify geriatric syndromes. Delirium in older people with cancer receiving palliative care arises from a complex interaction between predisposing and precipitating factors, making CGA useful for identifying the factors that present the greatest risk. The inpatients in this study had low functionality, nutritional status, and cognitive function, but adequate social support. Following CGA, variables such as male sex, dysphagia, and urinary catheterization were identified as risk factors. Identifying individuals at higher risk enables more comprehensive and personalized management. The prevention and effective management of delirium requires a person-centered, interdisciplinary approach that considers clinical, functional, and psychosocial aspects.

## Figures and Tables

**Table 1 nursrep-15-00328-t001:** Main characteristics of the sample.

	DG (n = 67) ^1^	NDG (n = 38) ^1^	Total (n = 105) ^1^	*p* Value
Age in years, mean (SD)	71.33 (4.24)	72.24 (3.74)	71.66 (4.1)	0.274 ^2^
Male	42 (62.7)	16 (42.1)	58 (55.2)	0.042 ^3^
Female	25 (37.3)	22 (57.9)	47 (44.8)	
Breast cancer	5 (7.4)	3 (7.8)	8 (7.7)	0.002 ^3^
Gastrointestinal cancer	27 (40.3)	6 (15.8)	33 (31.4)	
Gynecologic cancer	0 (0)	1 (2.6)	1 (0.9)	
Lung cancer	9 (13.4)	12 (31.6)	21 (20.1)	
Lymphoma	19 (28.3)	5 (13.1)	24 (22.8)	
Prostate cancer	4 (5.9)	2 (5.2)	6 (5.7)	
Skin cancer	3 (4.4))	9 (23.6)	12 (11.4)	

^1^ All data are presented as number (%) of participants unless specified otherwise. ^2^ Student’s *t*-test. ^3^ Chi-square test. Abbreviations: DG, delirium group; NDG, non-delirium group; SD, standard deviation.

**Table 2 nursrep-15-00328-t002:** Mean differences between groups in the comprehensive geriatric assessment.

	Mean (SD)		MD (95%CI)	*p* Value
	DG (n = 67)	NDG (n = 38)		
Barthel Index ^1^	22.99 (14.92)	25.26 (16.43)	2.27 (−3.96 to 8.50)	0.470 ^2^
MNA ^3^	16.80 (2.51)	17.14 (2.96)	0.34 (−0.74 to 1.42)	0.536 ^2^
MMSE ^4^	7.19 (2.36)	8.08 (2.92	0.88 (−0.16 to 1.93)	0.095 ^2^
GSFES ^5^	13.25 (2.45)	12.42 (2.59)	−0.83 (−1.84 to 1.77)	0.036 ^2^

^1^ 0–100, higher score = greater independence. ^2^ Student’s *t*-test. ^3^ 0–30, higher score = better nutritional status. ^4^ 0–30, higher score = better cognitive status. ^5^ 5–25, higher score = lower social risk. Abbreviations: CI, confidence interval; DG, delirium group; GFES, Gijon social-familial evaluation scale; MD, mean difference; MMSE, Mini Mental State Examination; MNA, Mini Nutritional Assessment; NDG, non-delirium group.

**Table 3 nursrep-15-00328-t003:** Risk factors for delirium.

Risk Factors	Number (%) of Participants		OR (95% CI)	*p* Value
	DG (n = 67)	NDG (n = 38)		
Age ≥ 75 years	15 (22.8)	9 (21.1)	1.08 (0.41–2.82)	0.559
Male sex	42 (40)	16 (15.23)	2.31 (1.02–5.20)	0.041
Dementia	30 (44.8)	18 (47.3)	0.90 (0.40–2.01)	0.798
Incontinence	52 (77.6)	26 (68.4)	1.60 (0.65–3.90)	0.305
Urinary catheterization	24 (35.8)	6 (15.8)	2.97 (1.09–8.13)	0.024
UTI in last 6 months	12 (17.9)	10 (26.3)	0.61 (0.23–1.58)	0.309
Kidney failure	4 (6)	4 (10.5)	0.54 (0.12–2.29)	0.477
Dehydration	6 (9)	1 (2.6)	3.63 (0.42–31.43)	0.181
Fall in previous 30 days	8 (11.9)	10 (26.3)	0.38 (0.13–1.06)	0.060
Dysphagia	29 (43.3)	9 (23.7)	2.45 (1.01–5.99)	0.041
Delirium in las 12 months	31 (46.3)	5 (13.2)	5.68 (1.97–16.34)	<0.001
Anticholinergics	18 (26.9)	11 (28.9)	0.90 (0.37–2.18)	0.819
Anxiolytics	10 (14.9)	5 (13.2)	1.15 (0.36–3.67)	0.803
Sedatives	44 (65.7)	30 (78.9)	0.52 (0.20–1.29)	0.145
Opioids	15 (22.4)	8 (21.1)	1.08 (0.41–2.85)	0.873
Polypharmacy ^1^	64 (95.5)	34 (89.5)	2.51 (0.53–11.89)	0.243
Antidepressants	25 (37.3)	22 (57.9)	0.81 (0.36–1.84)	0.629
Neuroleptics	46 (68.7)	30 (78.9)	0.58 (0.22–1.48)	0.251
Barthel Index < 60 points	64 (95.5)	36 (94.7)	1.18 (0.18–7.42)	0.857
MNA < 17 points	30 (44.8)	17 (44.7)	1.00 (0.45–2.23)	0.997
MMSE < 10 points	63 (94)	33 (86.8)	2.38 (0.60–9.49)	0.216
GFES ≥ 15 points	10 (14.9)	4 (10.5)	1.49 (0.43–5.12)	0.518

^1^ ≥5 prescribed drugs. Abbreviations: CI, confidence interval; DG, delirium group; GFES, Gijon social-familial evaluation scale; MMSE, Mini Mental State Examination; MNA, Mini Nutritional Assessment; NDG, non-delirium group; UTI, urinary tract infection.

**Table 4 nursrep-15-00328-t004:** Predictive model.

Variables	Odds Ratio	95% Confidence Interval	*p* Value
Male sex	2.216	0.953–5.152	0.065
Urinary catheterization	2.826	1.003–7.961	0.049
Dysphagia	2.283	0.907–5.746	0.080

## Data Availability

Data available on request from the authors.

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
