# Peer review of "Identifying Risk Factors for Delirium Through Comprehensive Geriatric Assessment in Older Adults Receiving Palliative Cancer Care"

_nursrep, 2025, doi:10.3390/nursrep15090328_

Round 1

Reviewer 1 Report

Comments and Suggestions for Authors

Thanks for giving me the opportunity to review this manuscript. Below are my comments to the authors:

Line 42, PMID: 31722766should be added to reference 4 as it is more specific

Line 42, the relation between chemobrain and delirium is barely studies. It would be a great addition to this paper to assess the relation between current and history for chemo brain and delirium and include it as a precipitating factor.

Line 59, please change cancer patients to "older patients diagnosed with advanced cancer and receiving palliative care..."

Line 78, What about cancer stage and site of metastasis "brain mets"?

Line 79, UTI right before delirium is a precipitating factor. But if several months before delirium, it might not be precipitating factor. In other word, a period of 6 month to link these factors with delirium is extremely wide. If not please provide a reference. Take "falls in the last month" as a good example  to avoid non-specificty of these factors.

Line 80, again specified time period (6 months) is very wide and concerning.

Line 81, authors should include those receiving sedatives such dexmedetomidine and ketamine

Table 2, MMSE score comparison. This indicates the possibility of chemobrain or brain mets in NDG which made their scores almost equal to the DG

Line 175. This should be a univariate analysis. Each risk or precipitating factor should be checked in DG vs. NDG individually

Line 177. Again NDG could have higher incidence of foggy brain (not dementia) unless clinically diagnosed and treated for dementia. Authors should investigate that and consider it in analysis if so

Line 185. Univariate analysis should be performed and shown to confirm these results.

Author Response

REVIEWER 1

Thanks for giving me the opportunity to review this manuscript. Below are my comments to the authors:

We would like to thank the reviewer for their valuable comments, which we have taken into account in this revised manuscript. Itemized responses are listed below. All the modifications have been marked in red throughout the manuscript to facilitate review.

(COMMENT 1) Line 42, PMID: 31722766should be added to reference 4 as it is more specific

Author’s answer 1: Thank you for the suggestion. The authors have added the reference.

(COMMENT 2) Line 42, the relation between chemobrain and delirium is barely studies. It would be a great addition to this paper to assess the relation between current and history for chemo brain and delirium and include it as a precipitating factor.

Author’s answer 2: Thank you for the suggestion, but the authors cannot perform this analysis because we did not collect this variable. The authors have added it in the limitations section (Lines 319-324).

(COMMENT 3) Line 59, please change cancer patients to "older patients diagnosed with advanced cancer and receiving palliative care..."

Author’s answer 3: Thank you for the suggestion. The authors have reworded the sentence according to the reviewer’s suggestion (lines 79-80).

(COMMENT 4) Line 78, What about cancer stage and site of metastasis "brain mets"?

Author’s answer 4: Thank you for the suggestion, but the authors cannot perform this analysis because we did not collect this variable. The authors have added it in the limitations section. We will take it into account for future studies (Lines 319-324).

(COMMENT 5) Line 79, UTI right before delirium is a precipitating factor. But if several months before delirium, it might not be precipitating factor. In other word, a period of 6 month to link these factors with delirium is extremely wide. If not please provide a reference. Take "falls in the last month" as a good example  to avoid non-specificty of these factors.

(COMMENT 6) Line 80, again specified time period (6 months) is very wide and concerning.

Author’s answer 5-6: Thank you for the suggestion, the authors have added the following reference and explanatory sentence:

Rodriguez-Mañas, L. Urinary Tract Infections in the Elderly: A Review of Disease Characteristics and Current Treatment Options. Drugs Context 2020, 9, 2020-4–13, doi:10.7573/dic.2020-4-13.

“Six months was considered an acceptable duration for evaluating the presence of urinary tract infection because older people often receive prolonged treatment for such infections.” (lines 103-105)

(COMMENT 7) Line 81, authors should include those receiving sedatives such dexmedetomidine and ketamine

Author’s answer 7: Thank you for the suggestion, the authors checked the records and added opioids, sedatives and polypharmacy (≥ 5 prescribed drugs) in the methods and results section, but there were no differences between groups (lines 102-103 and table 3).

(COMMENT 8) Table 2, MMSE score comparison. This indicates the possibility of chemobrain or brain mets in NDG which made their scores almost equal to the DG

Author’s answer 8: Thank you for the suggestion, the authors have discussed this idea in the discussion section (lines 266-274).

(COMMENT 9) Line 175. This should be a univariate analysis. Each risk or precipitating factor should be checked in DG vs. NDG individually

Line 185. Univariate analysis should be performed and shown to confirm these results.

Author’s answer 9: Thank you for these comments. Table two provides the frequencies of each factor in both groups (univariate analysis) as well as the association between each factor and the presence of delirium (the OR; bivariate analysis). We are unsure what the reviewer considers is missing from the table.

(COMMENT 10) Line 177. Again NDG could have higher incidence of foggy brain (not dementia) unless clinically diagnosed and treated for dementia. Authors should investigate that and consider it in analysis if so

Author’s answer 10: Thank you for the suggestion, the authors have discussed this idea in the discussion section (lines 266-274).

Reviewer 2 Report

Comments and Suggestions for Authors

Since delirium is highly prevalent in the context of palliative care and in cancer patients, and given its complexity and multifactorial causes, studies in this field are necessary and of scientific and clinical relevance. Despiste the relevance, there are some aspects that need improvement and correction in this manuscript.

Introduction:

- Since the cause of delirium is multifactorial, it will be necessary to frame the variables mentioned in the article title and justify the authors' choice of inclusion criteria. A more targeted literature review is therefore necessary.

Study Objectives:

- If the study objective is to identify risk factors for delirium, why the authors do specifically mention only those identified in the study results in the title? Other factors may exist and were not explored in this study. Wouldn't it be necessary for the objectives to be more specific to the variables under study? The title is not aligned with the objective.

- I also suggest aligning it with the discussion, where at the beginning they mention that the objective was to develop a model of the predictors of delirium in older hospitalized cancer patients receiving palliative care. I don't think the authors have developed a model.

Method:

- Observational longitudinal study. I do not find any monitoring over time in the results presented. Were these analyses performed? They are not explained and not even presented in the manuscript. Won't the study be cross-sectional?

- Clarification and coherence between the type of study and results presented is necessary.

Procedures:

- No patient was medicated with morphine? There is evidence in the literature that high doses of morphine are one of the most prevalent factors in the presence of delirium, in addition to other types of drugs.

- The procedures should be clear regarding the different time points at which the data were collected.

Results:

- It's unclear which variables were selected for regression analysis, regardless of the results.

- In the analyses, I suggest controlling medication using it as a covariate in the analyses.

- In lines 135 and 175, the word "bivariable" will not be "bivariate"?

Discussion:

The discussion will have to be modified considering the changes you make to the results section.

Author Response

REVIEWER 2

Since delirium is highly prevalent in the context of palliative care and in cancer patients, and given its complexity and multifactorial causes, studies in this field are necessary and of scientific and clinical relevance. Despiste the relevance, there are some aspects that need improvement and correction in this manuscript.

We would like to thank the reviewer for their valuable comments, which we have taken into account in this revised manuscript. Itemized responses are listed below. All the modifications have been marked in red throughout the manuscript to facilitate review.

(COMMENT 1) Introduction:

- Since the cause of delirium is multifactorial, it will be necessary to frame the variables mentioned in the article title and justify the authors' choice of inclusion criteria. A more targeted literature review is therefore necessary.

Author’s answer 1: Thank you for the suggestion. The authors have added more information in the Introduction section from lines 33 to 37 and from lines 67 to 76.

(COMMENT 2)Study Objectives:

- If the study objective is to identify risk factors for delirium, why the authors do specifically mention only those identified in the study results in the title? Other factors may exist and were not explored in this study. Wouldn't it be necessary for the objectives to be more specific to the variables under study? The title is not aligned with the objective.

- I also suggest aligning it with the discussion, where at the beginning they mention that the objective was to develop a model of the predictors of delirium in older hospitalized cancer patients receiving palliative care. I don't think the authors have developed a model.

Author’s answer 2: Thank you for the suggestion, the authors have reworded the justification (lines 67 to 80) and changed the title: “Identifying risk factors for delirium through comprehensive geriatric assessment in older adults receiving palliative cancer care”.

(COMMENT 3) Method:

- Observational longitudinal study. I do not find any monitoring over time in the results presented. Were these analyses performed? They are not explained and not even presented in the manuscript. Won't the study be cross-sectional?

- Clarification and coherence between the type of study and results presented is necessary.

Author’s answer 3: Thank you for this comment. Our study included longitudinal follow-up of all participants during their hospital stay. None had delirium at baseline (admission to hospital), and the nursing staff screened for delirium every day. We have added some clarifications in the methods section and have reported the mean length of hospital stay (follow-up) in the results.

(COMMENT 4) Procedures:

- No patient was medicated with morphine? There is evidence in the literature that high doses of morphine are one of the most prevalent factors in the presence of delirium, in addition to other types of drugs.

Author’s answer 4: Thank you for the suggestion, the authors checked the records and added opioids, sedatives and polypharmacy (≥ 5 prescribed drugs) in the methods and results section, but there were no differences between groups (lines 102-103 and table 3).

- (COMMENT 5) The procedures should be clear regarding the different time points at which the data were collected.

Author’s answer 5: Thank you for the suggestion, the authors have added more information in the Methods section.

Results:

-(COMMENT 6)  It's unclear which variables were selected for regression analysis, regardless of the results.  In the analyses, I suggest controlling medication using it as a covariate in the analyses.

Author’s answer 6 : Thank you for the suggestion, the authors have reworded the statistics section (lines 159-182).

- (COMMENT 7) In lines 135 and 175, the word "bivariable" will not be "bivariate"?

Author’s answer 7: Thank you for the suggestion, the authors have changed the word.

Discussion:

(COMMENT 8) The discussion will have to be modified considering the changes you make to the results section.

Author’s answer 8 : Thank you for the suggestion, the authors have reworded the discussion section.

Reviewer 3 Report

Comments and Suggestions for Authors

Dear authors,

I find this article interesting because understanding the main risk factors for delirium in older adults would allow for its prevention and avoidance of complications arising from this problem.

I would not include the conclusions in the title, in this case the risk factors, because I believe it may detract from the interest of reading the article or at least the abstract.

The abstract is well-structured, providing a comprehensive and detailed overview of the study.

The introduction is concise, although I believe it addresses the main aspects of delirium in older adults.

The objective of the study is clearly defined.

The methodology is appropriate and is described in detail to allow for reproduction, although the inclusion and exclusion criteria of the study should be specifically specified. The type of sampling used in this study should also be indicated. Furthermore, it would be helpful to briefly explain the items, dimensions or characteristics, and type of scale of the different questionnaires used to facilitate understanding for readers unfamiliar with them. The results are presented in detail and with several tables that facilitate their understanding.

Why are parametric (Student's t-test) and nonparametric (Mann-Whitney U) tests used to analyze the results of the study (Table 2)? Are the data from the analyzed questionnaires quantitative or qualitative ordinal? This aspect of the study should be clarified, specifically by indicating the type of measurement scale used by the questionnaires.

The discussion is detailed and adequately analyzes the results obtained, establishing relationships with previous studies that also address delirium in older adults, although I think the non-oncological intervention plan that could be proposed for these patients should be indicated more specifically (Lines 224-226). I also believe that the limitations of the study should include those derived from the type of sampling, since it is not only the sample size that can reduce its external validity.

The conclusions are consistent with the results obtained and adequately respond to the proposed objective. The references are accurate and up-to-date.

Kind regards.

Author Response

REVIEWER 3

Dear authors,

I find this article interesting because understanding the main risk factors for delirium in older adults would allow for its prevention and avoidance of complications arising from this problem. I would not include the conclusions in the title, in this case the risk factors, because I believe it may detract from the interest of reading the article or at least the abstract. The abstract is well-structured, providing a comprehensive and detailed overview of the study. The introduction is concise, although I believe it addresses the main aspects of delirium in older adults. The objective of the study is clearly defined.

We would like to thank the reviewer for their valuable comments, which we have taken into account in this revised manuscript. Itemized responses are listed below. All the modifications have been marked in red throughout the manuscript to facilitate review.

Author’s answer: Thank you for the suggestion regarding our title. The authors have changed the title to “Identifying risk factors for delirium through comprehensive geriatric assessment in older adults receiving palliative cancer care”.

(COMMENT 1)The methodology is appropriate and is described in detail to allow for reproduction, although the inclusion and exclusion criteria of the study should be specifically specified. The type of sampling used in this study should also be indicated. Furthermore, it would be helpful to briefly explain the items, dimensions or characteristics, and type of scale of the different questionnaires used to facilitate understanding for readers unfamiliar with them.

Author’s answer 1: Thank you for the suggestion, the authors have added more information in the Methods section.

(COMMENT 2)The results are presented in detail and with several tables that facilitate their understanding. Why are parametric (Student's t-test) and nonparametric (Mann-Whitney U) tests used to analyze the results of the study (Table 2)? Are the data from the analyzed questionnaires quantitative or qualitative ordinal? This aspect of the study should be clarified, specifically by indicating the type of measurement scale used by the questionnaires.

Author’s answer 2: Thank you for the suggestion, the authors have added more information in the Methods section. The questionnaires were treated as dichotomous quantitative and qualitative data. The distribution of Barthel Index scores for this sample is normal; it was an error to indicate in the table in its first version that it was a nonparametric test.

(COMMENT 3) The discussion is detailed and adequately analyzes the results obtained, establishing relationships with previous studies that also address delirium in older adults, although I think the non-oncological intervention plan that could be proposed for these patients should be indicated more specifically (Lines 224-226).

Author’s answer 3: Thank you for the suggestion, the authors have added more information (lines 281-282).

(COMMENT 4) I also believe that the limitations of the study should include those derived from the type of sampling, since it is not only the sample size that can reduce its external validity.

Author’s answer 4: Thank you for the suggestion, the authors have added the type of sampling as a limitation (lines 319-320).

The conclusions are consistent with the results obtained and adequately respond to the proposed objective. The references are accurate and up-to-date.

Kind regards.

Round 2

Reviewer 1 Report

Comments and Suggestions for Authors

I would like to thank the authors for addressing my comments. I have no more comments to add.

Author Response

Thank you

Reviewer 2 Report

Comments and Suggestions for Authors

I thank the authors for addressing the suggested improvements and making the necessary modifications. I consider the article ready for publication. 

Author Response

Thank you

Reviewer 3 Report

Comments and Suggestions for Authors

Dear Authors,

I consider that the manuscript has been sufficiently improved and I have no additional comments or suggestions.

Kind regards.

Author Response

Thank you